# The Role of Self-Esteem in the Academic Performance of Rural Students in China

**DOI:** 10.3390/ijerph192013317

**Published:** 2022-10-15

**Authors:** Wenjing Yu, Yiwei Qian, Cody Abbey, Huan Wang, Scott Rozelle, Lauren Ann Stoffel, Chenxu Dai

**Affiliations:** 1China Academy for Rural Development, Zhejiang University Zijingang Campus, Hangzhou 310058, China; 2Stanford Center on China’s Economy and Institutions, Stanford University, Stanford, CA 94305, USA; 3Research Institute of Economics and Management, Southwestern University of Finance and Economics, Chengdu 610074, China

**Keywords:** self-esteem, academic performance, developing countries, rural China

## Abstract

The self-esteem of students may be significantly associated with their academic performance. However, past research in developing contexts on this issue is limited, particularly among early adolescents. Using a sample of 3101 students from rural primary and junior high schools in China, this study measured their self-esteem by the Rosenberg Self-Esteem Scale (RSES) and explored its association with academic performance. Our findings indicate that students in rural China had both significantly lower self-esteem and a higher prevalence of low self-esteem when compared to past studies of similarly aged students both from urban China and internationally. Furthermore, there was a strong positive correlation between a student’s self-esteem and academic performance. A one-SD increase in RSES score (indicating better self-esteem) was associated with an increase of 0.12 SD in standardized math scores (*p* < 0.001), and students with low self-esteem (RSES score < 25) scored lower on math tests by 0.14 SD (*p* < 0.001), which were robust and consistent when employing the propensity score matching method. Our study expands the growing body of empirical evidence on the link between self-esteem and academic performance among rural youth in developing countries and emphasizes the need to improve their self-esteem with the aim of helping them achieve academically.

## 1. Introduction

A recent wave of the literature in economics and psychology has established that individual attributes, such as personality, perseverance, dependability, and charisma, matters for later life success [1,2]. Those individual attributes, commonly noted as non-cognitive skills, has been documented as a powerful predictor of wages, schooling, health, and success in many other aspects of economic and social life [3,4,5].

The importance of non-cognitive skills in explaining later life outcome could stem from its complementarity with cognitive skills. Both theoretical and empirical research has formalized the strong link between non-cognitive skills and cognitive skills [6,7,8]. However, many different traits are usually lumped together as non-cognitive skills in the literature. As reliable measures of non-cognitive traits are becoming available, more efforts are needed to examine the role of various non-cognitive traits in skill development [5].

Self-esteem is one non-cognitive trait that is gaining attention for being one of the key determinants shaping academic achievement, which is an indicator of cognitive development [9,10]. Specifically, most educators believe that self-esteem is necessary for students to take risks in their learning, recover from adversity, and develop high perceptions of competence in specific academic subjects [11]. The theory is supported by many empirical findings from developed contexts. Earlier studies investigated that self-esteem is positive prospective related to success and schooling [12]. Substantial studies also indicate that, in general, children and adolescents with low self-esteem are more likely to achieve poor schooling, as they have relatively less motivation to learn [13], poorer logical ability [14], more distraction [15], higher school dropout rate [16], and lower learning efficiency [17].

Although a significant correlation between academic performance and self-esteem has been highlighted by studies conducted in well-off nations, such as in Saudi Arabia, United States, and England [18,19], the evidence from developing contexts (i.e., low- and middle-income countries (LMICs)) is still limited. In addition, there is an urgent need to pay more attention to the students in economically disadvantaged areas, as they are more likely to have relatively poor self-esteem [20,21,22]. Current studies in LMICs is mainly focus on high school and college youths [23,24], with limited empirical evidence about students in elementary and junior high school [25], despite the importance of early adolescence as a stage during one’s development [26]. Besides, most existing assessments take into account the individual characteristics of students [20,27], but they seldom account for their family and social environment. Since family background and social environment are closely related to individual academic abilities, failing to control for them could lead to a biased association between self-esteem and academic achievement [28]. Overall, the relationship between academic performance and self-esteem across developing contexts remains understudied. 

Among developing contexts, rural China is an ideal setting to study this issue. The socioeconomic status of China’s rural population lags far behind that of its urban residents [29,30]. Approximately 70% of students in China reside in rural areas [31]. Due to a combination of factors including high rates of parental migration and educational resource gaps, rural students in China generally have lower levels of social support both inside and outside the home [32], and they also have lower quality educational resources [33]. Thus, they are more likely to have poor academic competence [34,35] and delayed psychological development [36,37]. Despite most researchers favoring the hypothesis of a link between self-esteem and academic performance in rural China, few empirical studies have explored this association [33,38,39], and the evidence collected from the general population of primary and lower secondary students was limited. Hence, understanding the relationship between self-esteem and academic performance among rural youth is critical, as it can contribute to creating evidence-based interventions that increase the academic outcomes in rural China. 

To fill these gaps in the literature, our present study focuses on unveiling the relationship between self-esteem and academic performance among children and adolescents in rural China. Our specific objectives are to (a) measure the self-esteem of a large sample of children and adolescents in rural China using standardized and internationally validated scales; (b) identify the social-environmental factors and students’ behavioral characteristics associated with student self-esteem and their association with the likelihood that students suffer from low self-esteem; (c) measure the association between student self-esteem and academic performance.

## 2. Materials and Methods

To achieve our objectives, we conduct this study using cross-sectional data collected from 30 rural schools in Lingtai County in the northwestern province of Gansu, China, where around 48% of the population are rural residents, compared to 36% in China’s overall population [31]. In Lingtai County, where the sample is located, the yearly income of rural households per capita (1622 USD) is significantly lower than the national average income of rural Chinese residents (2693 USD) [40] and lies in the second-lowest income quintile among the rural population [31]. 

### 2.1. Ethical Review

The Institutional Review Board (IRB) of Stanford University approved this study (Protocol 58251). Caregivers of eligible students received written consent forms prior to the implementation of the study. We deleted student names from all electronic files during data encryption and forbade discussion about student answers both during and after the survey to comply with the Declaration of Helsinki and ensure the confidentiality of each student’s information. 

### 2.2. Sample Selection

To select our study sample, we used the following protocol. Thirty schools were selected (20 primary schools and 10 junior high schools) randomly from a list of all schools from the local education bureau, which were needed to reach 80% statistical power based on power calculation. We included fourth and fifth grade students in the selected primary schools and seventh and eighth grade students in the selected junior high schools. In each grade, we could only randomly select at most two classes. Therefore, if a grade only consisted of two or fewer classes, then all classes in the grade were chosen; if a grade consisted of more than two classes, then we randomly selected two classes. All students presented on the day of the survey in each sample class were selected to take the survey without compensation. Ultimately, we sampled 3101 students in 95 classes from 30 schools.

### 2.3. Major Blocks of Data

We collected three blocks of data for each of the participating households. These three blocks included student self-esteem (Block A), student academic performance (Block B); and sociodemographic and behavioral characteristics (Block C). In addition, we administered a set of questions related to basic sociodemographic and behavioral characteristics of students and their parents (Block C). Below, we describe the measurement tools and survey forms used for each block in detail.

Data Block A: Self-esteem status. The self-esteem of students was measured by the 10-item Rosenberg self-esteem scale (RSES), which is considered the gold standard of self-esteem measurement [40] and has been widely used to measure self-esteem in many countries both in developed and developing contexts [41]. Past studies that use the Chinese version of the RSES (used in this study) have demonstrated its reliability and validity [42,43,44]. The respondent was invited to answer each of the 10 items on a Likert scale of 1–4: 1 (“strongly disagree”), 2 (“disagree”), 3 (“agree”), 4 (“strongly agree”). After reversing the score for the five items that were negatively valenced (items 3, 5, 8, 9 and 10) and adding a positive score for the five positively valenced items (items 1, 2, 4, 6, and 7), a total score for the scale was created by adding the numerical value of the responses, resulting in scores ranging from 10 to 40, with higher scores indicating greater self-esteem and lower scores indicating lower self-esteem [45]. 

To the best of the our knowledge, there are no standard cut-off points for low self-esteem on the RSES [46]. However, almost all studies have adopted the practice of using the 25-point cut-off [47,48], which has been proved to have satisfactory discrimination to classify low self-esteem on the RSES [49]. Hence, the dummy variable was also defined in our study as “low self-esteem” if the RSES score was lower than 25 points.

Data Block B: Academic performance. Student academic performance was measured by a 30-min standardized math test with 30 multiple-choice items. Each item on the test is assigned a score of 1 if the answer is correct or 0 if it is incorrect, and the total test score is added up to 0–30. We chose math as the measurement of academic performance because achievement in math more directly relates to learning experiences at school compared to achievement in language and reading [50]. These math exams have been previously used by other research teams working to examine student academic performance in other parts of rural China, and the results are believed to be credible. For the current study, we also pre-tested the exams several times in sample schools to certify their relevance. To guarantee alignment with the national curriculum, educators from the local education bureau helped create the test items for each sample grade. The exam was strictly timed and administered by enumerators at each sample school, and test scores were later normalized according to the distribution of scores in each grade. 

Data Block C: Social-environmental and behavioral characteristics. We also collected sampling data on significant self-reported covariates: basic demographic characteristics of the student, including student gender (male or female), age (divided into two subgroups according to the distribution of the sample: ≤11, >11), and whether students were boarding at school (yes or no). We also measured the various parental characteristics: age of each student’s parents (years) and parental education level (with junior high school graduation as a cutoff) (>9 years education, yes or no). Students were categorized as “left-behind children” if both parents had migrated out for work for more than six months in the past year. 

Students were also asked to note the number of minutes spent after school on recreational reading (daily reading time over 30 min as the cutoff) and their phone (daily phone time over 30 min as the cutoff). Responses for how often a student typically engaged in group-based activities organized by their school were also coded into a dummy variable indicating whether the students often chose to participate in the activities.

### 2.4. Statistical Analysis

We estimate the association between the self-esteem and academic performance among the students in western rural China. The statistical analysis in our study followed four steps.

First, for the descriptive analysis, the summary statistics of the sample are reported, as well as the means, standard deviations (SD), and minimum and maximum value of students’ self-esteem calculated by RSES and academic performance. 

Second, we conducted *t*-tests to identify which sociodemographic and behavioral characteristics were associated with levels of self-esteem and academic performance. We also use regression analysis to examine the correlation between students’ self-esteem and these demographic and familial factors. An ordinary least squares (OLS) linear regression model was used to conduct the multivariate analysis to identify the sociodemographic and behavioral characteristics associated with students’ self-esteem. For binary outcomes, a probit model was also used to measure the association between control variables and the likelihood of students suffering from low self-esteem. 

Third, the association between student self-esteem and academic performance was measured while controlling for sociodemographic and behavioral characteristics with an OLS linear regression model. Considering the nested nature of the data, we cluster all standard errors at the class level. 

Fourth, besides controlling for sociodemographic and behavioral characteristics, we also adopt propensity score matching (PSM) method when estimating the correlation between self-esteem and academic performance. Following a series of well-established steps with common support checking and diversified matching approaches, we reduce the bias due to confounding variables to show the robust correlation between self-esteem on academic performance. 

We conducted all analyses in Stata 16.1 (StataCorp LP, College Station, TX, USA). *p*-values below 0.05 were deemed as statistically significant.

## 3. Results

### 3.1. Descriptive Statistics

Table 1 shows the main summary statistics of our sample. 53.8% of the sample is male. The average age was 11.5 years old, and more than half of the students (60.3%) were elementary school students. Among them, a small portion of the participants boarded at school (14.6%). Most of the students’ fathers were migrants (56.2%), compared to only about one quarter of mothers (26.4%). One-fifth of the children were left-behind children (20.2%), meaning that their parents were both migrants. 

The parents of our sample were generally in working age, and the mothers were slightly younger than the fathers, with an average age of 38 and 41 years, respectively. However, parental education level was relatively low on average. Only a few fathers (23.7%) and mothers (14.4%) had a junior high school education or above, which was far lower than the overall share of adults in China who had attended junior high school (65.1%) and even lower than that in Gansu Province overall (54.9%) (National Bureau of Statistics of China, 2021).

Table 1 reports the mean self-esteem scores of the sample, as measured by the RSES. The mean RSES score of sampled students was 26.54, with a standard deviation of 4.28. Almost a third of the students (30.1%) had low self-esteem problem, as indicated by the share of students with RSES scores lower than 25 points. 

We also report summary statistics of a few key after-school behaviors of students. Specifically, the average daily recreational reading time after school was 27 min, and about 55% of the students spent more than 30 min on reading recreationally. Meanwhile, sample students spent an average of 10 min using phones each day after school, and there was a small percentage of students (14%) that spent more than 30 min on their phones. Finally, more than two-thirds of students indicated that they often attended group activities at school. 

### 3.2. Self-Esteem and Academic Performance: Comparisons between Different Subgroups of Students

Table 2 explores the differences in RSES scores and the shares of students with low self-esteem between subgroups. Students from junior high school had a significantly higher mean RSES score than those in elementary school (0.45-point difference, *p* = 0.004). Those who did over 30 min of recreational reading after school (0.59-point difference, *p* < 0.001), those who spent less than 30 min on the phone every day (0.71-point difference, *p* = 0.001), and those who often attended group activities at school (1.09-point difference, *p* < 0.001) not only demonstrated a significantly higher average mean RSES score than their peers, but also were significantly less likely to have low self-esteem, with differences of 3.9% (*p* = 0.018), 7.4% (*p* = 0.002), and 7.8% (*p* < 0.001), respectively.

We also find a strong association between parental factors and student self-esteem. On average, those students whose fathers had been educated for more than nine years had a 0.63-point higher RSES score (*p* < 0.001) than their peers, while the RSES score for students whose mothers had at least nine years of education was 0.81-points higher (*p* < 0.001), in addition to a 6.7% (*p* = 0.004) lower risk of having low self-esteem. Likewise, students who had migrant fathers (0.42-point difference, *p* = 0.007) or migrant mothers (0.79-point difference, *p* < 0.001) both had a higher risk of suffering from low self-esteem (a 4.4% difference and a 6.8% difference, respectively, *p* < 0.01) than their peers. Furthermore, such gaps were larger between students whose parents were both migrants (LBCs) and their non-LBC peers (0.81-points lower, *p* < 0.001; and 6.7%, *p* = 0.001). Additionally, students in families with higher asset index scores were found to have a significantly higher mean RSES score (0.59–point difference, *p* < 0.001) and were less likely to have low self-esteem than their peers (4% difference, *p* = 0.036). There were also several subgroups of students that did not significantly differ by either mean RSES score or share of students with low self-esteem. These included gender, boarding status, and ethnicity.

### 3.3. OLS Regression of Factors Associated with Student Self-Esteem

Table 3 displays the results of the OLS regression analysis examining associations between social-environmental characteristics and self-esteem level. The first column displays the associations between social-environmental characteristics and student RSES score. The RSES scores of students whose mothers had higher education levels were 0.62 points higher than those of their peers on average (*p* = 0.013), while the scores of students whose mothers migrated were 0.67 points lower (*p* < 0.001). Meanwhile, students who read for more than 30 min a day (0.40-points, *p* = 0.011), who engaged in phone usage of less than a half hour (0.72-points, *p* = 0.002), and who were actively involved in group activities at school (1.01-points, *p* < 0.001) had significantly higher RSES scores.

The second and third columns in Table 3 show the association between social-environmental factors and low self-esteem. Maternal education level was negatively associated with low self-esteem (margins = 7%, *p* = 0.005). Similarly, students were more likely to have low self-esteem if their mother migrated compared to those whose mother stayed at home (margins = 6%, *p* = 0.003). Using a phone for more than half an hour after school or infrequently attending group activities were both significantly associated with having low self-esteem (margins = 7%, *p* = 0.002 and margins = 7%, *p* < 0.001, respectively).

In addition, a number of social-environmental characteristics were not associated with student self-esteem according to our adjusted model results (Table 3), either in terms of scores or probabilities. These characteristics include the student’s gender, age, ethnicity, boarding status, paternal education, paternal migration, and family assets.

### 3.4. OLS Regression of Student Self-Esteem and Academic Performance

We report the association between student self-esteem and academic performance in Table 4. We find significant associations between student self-esteem measured by RSES and their academic achievement. First, when measured as a continuous variable, a one-standardized deviation increase in student RSES score was associated with a 0.12 SD increase in standardized math score (*p* < 0.001) (Table 4 column 2), when controlling for all parental, familial, and social-environmental characteristics listed in Table 2. Second, we find that students suffering from low self-esteem had significantly lower math scores (−0.14 SD, *p* < 0.001) (Table 4 column 4). Both when using continuous or dummy measures of self-esteem, we found student self-esteem have positive correlation with academic performance.

### 3.5. Robust Check Using Propensity Score Matching Methods

To reduce the bias due to confounding factors in the OLS estimation, we used the propensity score matching method to examine the association between student self-esteem and academic performance. Following the steps to implement the matching estimator, we find that there are sufficiently large overlaps in the propensity scores of the treatment and control groups in our sample (Figure 1). Additionally, we use balance tests to check the distribution of the relevant covariates in both the treatment and comparison groups for each matching method. The results show that the treatment and comparison groups are all balanced for each form of PSM employed in our study (Figure 2).

The main results from the three different types of PSM analysis provide preliminary evidence that low self-esteem may negatively impact students’ educational outcomes (Table 5). Using kernel matching, we find that low self-esteem decreased student academic achievement by 0.13 SD (*p* < 0.01) (Row 1). Similarly, when using nearest neighbor matching (Row 2), the coefficient was slightly larger, resulting in a decrease of 0.14 SD (*p* < 0.05). In addition, when radius matching is employed (Row 3), we find that academic performance decreased by 0.169 SD (*p* < 0.001). In total, the results reached by different matching strategies are consistent in direction and have a similar magnitude; that is, self-esteem is positively related with academic performance (while low self-esteem is negatively related with academic performance). 

## 4. Discussion

This study empirically explores the association between the self-esteem and academic performance among students in rural China. First, using a data set that contains information on 3101 students and their parents, our study shows that the self-esteem is correlated with sociodemographic and behavioral characteristics, indicating parental roles and behaviors are related to the children’s self-esteem (Table 2 and Table 3). Second, we present evidence that self-esteem is strongly associated with students’ academic performance and that poor self-esteem may inhibit students’ learning potential (Table 4 and Table 5). Our results are robust when controlling for a range of basic characteristics, and it is also further strengthened by the PSM method. 

Our findings call for more attention to be paid to self-esteem on students in rural China. Compared to our results (mean RSES score in our sample = 26.53) (Table 1), the findings of most studies taking place both inside and outside of China that have used RSES among similarly aged students have reported higher self-esteem scores, including Guangzhou city in China (28.86) [51], Costa Rica (31.49) [51], and Eritrea (31.16) [52], though these cross-national differences may be partially mediated by cultural variations [41,53]. Nearly one-third of students in our sample have low self-esteem, which is also more severe than in other contexts, such as Vietnam (19.4%) [54] and Saudi Arabia (23.4%) [55]. Such a difference was also pointed out by earlier studies that economic conditions would have a positive correlation with self-esteem of young adolescents [56]. Students in any given society with high per-capita income and low unemployment rate might be more optimistic and hopeful of obtaining a secure job after their graduation, leading to a high quality of life, and this may enhance their self-esteem [57]. 

Our results add to emerging evidence from developing contexts about the importance of self-esteem as determinants of young adolescents’ academic outcomes. The evidence in Table 4 and Table 5 shows that higher levels of self-esteem are strongly and robustly associated with better academic performance among young adolescents, and that students suffering from low self-esteem (beneath the cut-off) were more likely to obtain lower math scores. It is consistent with the other study saying that self-esteem serves as a precursor of educational attainment [58]. It is also possible, however, that higher academic performance is influencing self-esteem, and their relationship may be bidirectional [59], as students with higher cognitive abilities and superior academic results seem more likely to reflect on and be aware of their own thoughts and feelings [60]. While our results do not establish a causal relationship, our present findings show that self-esteem has a significant and robust correlation with the cognitive ability development in school-age children in economically disadvantaged regions. Considering the low levels of self-esteem among students in rural China and the importance of cognitive development, our results call for future study to investigate the potentially substantial impact of non-cognitive skills on cognitive development.

This is consistent with the expectations of economics theory that cognitive and noncognitive abilities are complements [2,61], so that under the premise that individuals are uncertain about their own ability, higher self-esteem enhances one’s effort toward achieving goals and ultimately leads to better outcomes [62]. Earlier empirical research has provided evidence that there is a strong positive relationship between non-cognitive abilities (i.e., self-esteem) and cognitive skills development (i.e., academic outcomes). For example, a wealth of correlational research has examined those higher levels of academic self-concept are associated with higher levels of achievement [22,44]. One meta-analysis including studies conducted among the formal school students (preschool to secondary) reported the average correlation between global self-concept and academic achievement was between 0.12 and 0.27 [63], while another study reported the results among adolescents populations (age 11 through college freshmen) to be even higher (0.31) [64]. Their findings also indicate that children with higher self-esteem are more likely to have better academic performance at their schooling age, suggesting non-cognitive development begets cognitive development. 

Additionally, this is one of the first studies to examine associations between early adolescent self-esteem and a variety of social-environmental factors in developing contexts. Our findings show that maternal education level is positively related to students’ self-esteem, while maternal migration is related to a higher risk of being low self-esteem, which has also been reported in past literature [65,66]. However, not all of our findings were consistent with past studies, as there were also no significant associations between self-esteem and certain variables found previously to be important in other contexts, including paternal migration [65] and education level [66]. Parenting style is closely linked to children’s psychological problems [67] and past research in rural China has also generally indicated maternal migration has a stronger impact on child mental health, including their self-esteem [68,69], which is perhaps due to their higher engagement in child rearing and interaction with their kids [65,70]. In general, findings in middle-income contexts show that fathers are playing increasingly important roles in children’s development, especially as fathers’ participation in child rearing has been increasing in recent years [71,72,73]. Thus, it is possible that in our study there may be other factors influencing these outcomes such as respondent age, survey time, and methods of survey administration, as well as context-specific factors such as social determinants of health. Further research is necessary to better understand how paternal and maternal migration differentially affect child development.

In addition to the social environmental factors, we also demonstrate that students’ behavioral characteristics are associated with their self-esteem. Less screen time, more recreational reading, and more involvement in group activities are all positively related to students’ self-esteem, which is consistent with past studies [74,75,76]. It is possible that recreational reading, especially when involving quality books, may support students to efficiently express their feelings and thoughts by enhancing their comprehension level, which can lead students to become more sociable and more self-confident [77]. Furthermore, involvement in extracurricular activities provides adolescents with a place to establish their own identities, as well as a context for self-assessment outside the more restricted expectations of school and family settings [78]. These findings indicate that in rural China and other developing contexts, one possible way to improve students’ academic performance is to enhance their self-esteem. In addition, self-esteem correlates with students’ social-environmental conditions and several behavioral characteristics in the education process. However, our results do not establish a causal relationship, more empirical research is needed to investigate which intervention designs are most effective at promoting the self-esteem of rural students in China, as well as which students (i.e., those of different genders, ages, etc.) benefit more from which types of designs. 

We note several limitations of this study. First, as this study used a cross-sectional and non-experimental design, the interpretation of the results needs to proceed with caution. For this reason, future research using data from randomized control trials or longitudinal studies is in need to determine the causal relations between self-esteem and students’ academic achievement. Second, the results may not apply to other contexts outside of less developed areas of rural China, as the adversities that students face as well as the socio-environmental determinants of self-esteem may differ across contexts. Finally, we did not have parent- or teacher-report of student mental or behavioral health in our current study. Accompanying student self-report measures of self-esteem with reports by the adults in their lives may strengthen the accuracy of measurement. Relatedly, the psychological conditions of parents and teachers might also be a social-environmental factor related to the development of the children and adolescents [79]. Thus, future studies should consider including measures of parents’ and teachers’ own well-being as potential social-environmental factors that are linked with student self-esteem. 

## 5. Conclusions

Using a sample of 3101 students selected from 30 primary and junior high schools in a rural region of northwestern China, we measured student self-esteem using an internationally validated scale (RSES), identified social-environmental correlates of self-esteem, and measured the robust correlation of self-esteem and academic performance with both OLS and PSM. Our results show that sample students had lower self-esteem compared to most past studies inside and outside of China and that there was a high prevalence of low self-esteem among primary and junior high school students in rural China. A number of social-environmental protective factors (maternal education level, frequent involvement in group activities, recreational reading) and risk factors (screen time and maternal migration) were also significantly associated with children’s self-esteem. Our main finding was that there was a strong association between self-esteem and academic performance; specifically, a one-point increase in the RSES score was significantly associated with a 0.03 SD higher math score, and having low self-esteem was associated with a significant 0.14 SD lower math score. The PSM method further indicated the robustness of the analyses. 

This study adds to the existing literature on the links between student’s self-esteem and academic performance of early adolescents in the context of developing countries. Our results highlight that the self-esteem of students in less developed, rural areas of China needs more attention from policymakers hoping to improve educational outcomes in rural China, and these findings provide insight into the role that interventions designed to enhance self-esteem could play in improving the academic performance of students in underdeveloped areas. Future studies with longitudinal research designs should also consider the determinants of self-esteem across multiple dimensions to obtain a better understanding of how to improve student self-esteem, as well as what factors mediate the relationship between cognitive and noncognitive skills.

## Figures and Tables

**Figure 1 ijerph-19-13317-f001:**
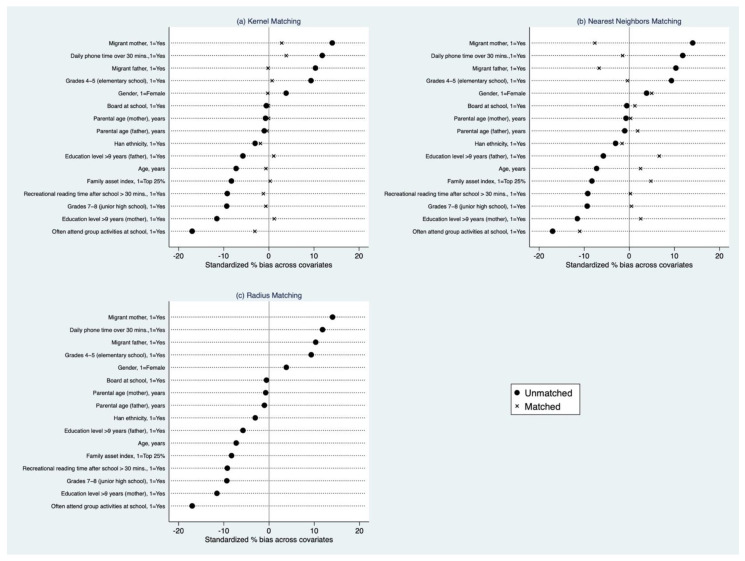
Standardized percentage bias across covariates after matching with multiple strategies. (**a**) the result with kernel matching; (**b**) the result with nearest neighbors matching; (**c**) the result with radius matching.

**Figure 2 ijerph-19-13317-f002:**
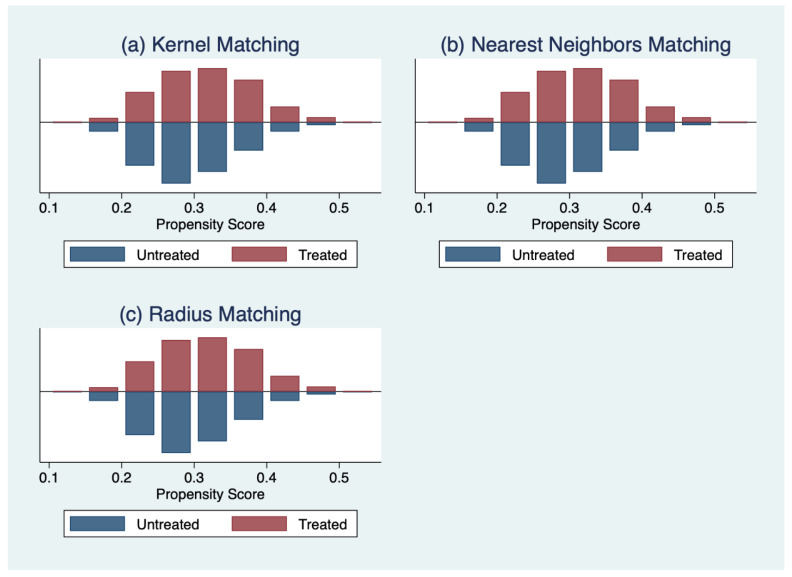
Overlap in the support of covariates by multiple matching strategies. (**a**) the result with kernel matching; (**b**) the result with nearest neighbors matching; (**c**) the result with radius matching.

**Table 1 ijerph-19-13317-t001:** Summary statistics.

	Num.	Mean/Percent	Std. Dev.	Min	Max
Rosenberg Self-Esteem Scale (RSES) Score	3101	26.54	4.284	11	40
Low self-esteem, 1 = Yes	3101	30.1%			
Gender, 1 = Female	3101	46.1%			
Age, years	3101	11.51	1.633	7.833	15.583
Grades 4–5 (elementary school), 1 = Yes	3101	60.27%			
Grades 7–8 (junior high school), 1 = Yes	3101	39.73%			
Board at school, 1 = Yes	3101	14.6%			
Father’s age, years	3101	41.05	6.157	23	71
Mother’s age, years	3101	38.13	5.701	21	66
Father’s education level (>9 years), 1 = Yes	3101	23.7%			
Mother’s education level (>9 years), 1 = Yes	3101	14.4%			
Left-behind child (both parents are migrants), 1 = Yes	3101	20.2%			
Migrant father, 1 = Yes	3101	56.2%			
Migrant mother, 1 = Yes	3101	26.4%			
Standardized values of math test score	3100	0.001	0.997	−4.573	1.902
Family asset index	3100	0	1.245	−2.241	2.909
Recreational reading time after school, mins	3101	27.53	24.426	0	700
Reading time over 30 min, 1 = Yes	3101	54.7%			
Phone time after school, mins	3101	10.37	23.268	0	600
Daily phone time over 30 min, 1 = Yes	3101	13.9%			
Often attend group activities at school, 1 = Yes	3101	69.1%			

**Table 2 ijerph-19-13317-t002:** The differences in mean RSES scores and low self-esteem percentage between subgroups of students.

Characteristics	Percent of Sample	RSES Score	Difference	Low Self-Esteem, 1 = Yes	Difference
Total	100	26.537		30.1%	
Gender					
Female	46	26.425	−0.209	0.31	−0.016
Male	54	26.634		0.293	
Grade					
Grades 4–5 (elementary school)	60	26.357	−0.454 **	0.317	0.040 *
Grades 7–8 (junior high school)	40	26.811		0.277	
Board at school					
Yes	15	26.625	0.102	0.298	−0.003
No	85	26.522		0.301	
Han ethnicity					
Yes	96	26.555	0.472	0.299	−0.034
No	4	26.083		0.333	
Father’s education level					
>9 years	24	27.015	0.626 ***	0.279	−0.028
≤9 years	76	26.389		0.308	
Mother’s education level					
>9 years	14	27.23	0.810 ***	0.243	−0.067 **
≤9 years	86	26.42		0.311	
Left-behind children					
Yes	20	25.893	−0.807 ***	0.354	0.067 **
No	80	26.7		0.287	
Migrant father					
Yes	56	26.353	−0.421**	0.32	0.044 **
No	44	26.774		0.276	
Migrant mother					
Yes	26	25.957	−0.788 ***	0.351	0.068 ***
No	74	26.745		0.283	
Family assets in top 25% of sample					
Yes	25	26.978	0.588 ***	0.271	−0.040 *
No	75	26.390		0.311	
Recreational reading time after school over 30 min					
Yes	55	26.805	0.590 ***	0.283	−0.039 *
No	45	26.215		0.322	
Daily phone time over 30 min					
Yes	14	25.923	−0.713 **	0.364	0.074 **
No	86	26.636		0.291	
Often attend group activities at school					
Yes	69	26.876	1.098 ***	0.277	−0.078 ***
No	31	25.779		0.355	

Note: *** *p* < 0.001, ** *p* < 0.01, * *p* < 0.05.

**Table 3 ijerph-19-13317-t003:** Correlation between individual and household characteristics and self-esteem level.

Individual and Household Characteristics	RSES Score	Low Self-Esteem, 1 = Yes	
	OLS	Probit Model	Marginal Effects
	(1)	(2)	(3)
Gender, 1 = Female	−0.28	0.08	0.03
	(0.15)	(0.05)	(0.02)
Age, years	−0.22	0.06	0.02
	(0.13)	(0.04)	(0.01)
Board at school, 1 = Yes	−0.31	0.16	0.05
	(0.28)	(0.09)	(0.03)
Han ethnicity, 1 = Yes	0.13	−0.04	−0.01
	(0.41)	(0.13)	(0.04)
Father’s age, years	−0.01	−0.00	−0.00
	(0.02)	(0.01)	(0.00)
Mother’s age, years	0.00	0.00	0.00
	(0.02)	(0.01)	(0.00)
Father’s education level (>9 years), 1 = Yes	0.31	0.02	0.01
	(0.20)	(0.07)	(0.02)
Mother’s education level (>9 years), 1 = Yes	0.62*	−0.23 **	−0.07 **
	(0.25)	(0.08)	(0.03)
Migrant father, 1 = Yes	−0.17	0.07	0.02
	(0.16)	(0.05)	(0.02)
Migrant mother, 1 = Yes	−0.67 ***	0.17 **	0.06 **
	(0.18)	(0.06)	(0.02)
Family asset index, 1 = Top 25%	0.33	−0.03	−0.01
	(0.21)	(0.07)	(0.02)
Recreational reading time after school > 30 min, 1 = Yes	0.40 *	−0.09	−0.03
	(0.16)	(0.05)	(0.02)
Daily phone time over 30 min, 1 = Yes	−0.72 **	0.22 **	0.07 **
Often attend group activities at school, 1 = Yes	1.01 ***	−0.22 ***	−0.07 ***
	(0.17)	(0.05)	(0.02)
Fixed effect	Yes	Yes	
Constant	27.76 ***	−0.90	
	(1.56)	(0.51)	
Observations	3101	3101	3101
R-squared	0.091		

Note: Standard errors clutered at the class level are in parentheses. *** *p* < 0.001, ** *p* < 0.01, * *p* < 0.05.

**Table 4 ijerph-19-13317-t004:** Correlation between self-esteem and student academic achievement.

Individual and Household Characteristics	Standardized Math Score			
	(1)	(2)	(3)	(4)
Standardized RSES score	0.14 ***	0.12 ***		
	(0.02)	(0.02)		
Low self-esteem, 1 = Yes			−0.18 ***	−0.14 ***
			(0.04)	(0.04)
Gender, 1 = female		−0.08 **		−0.09 **
		(0.03)		(0.03)
Age, years		−0.06 **		−0.06 **
		(0.03)		(0.03)
Board at school, 1 = Yes		−0.09		−0.09
		(0.06)		(0.06)
Han ethnicity, 1 = Yes		0.17 *		0.17*
		(0.09)		(0.09)
Father’s age, years		−0.01 ***		−0.01 ***
		(0.00)		(0.00)
Mother’s age, years		0.01 **		0.01 **
		(0.01)		(0.01)
Father’s education level (>9 years), 1 = Yes		0.09 **		0.10 **
		(0.04)		(0.05)
Mother’s education level (>9 years), 1 = Yes		0.06		0.07
		(0.06)		(0.06)
Migrant father, 1 = Yes		0.12 ***		0.12 ***
		(0.04)		(0.04)
Migrant mother, 1 = Yes		−0.02		−0.03
		(0.04)		(0.04)
Family asset index, 1 = Top 25%		0.08 *		0.09 *
		(0.05)		(0.05)
Recreational reading time after school over 30 min, 1 = Yes		0.21 ***		0.22 ***
		(0.04)		(0.04)
Daily phone time over 30 min, 1 = Yes		−0.07		−0.08
		(0.05)		(0.05)
Often attend group activities at school, 1 = Yes		0.15 ***		0.17 ***
		(0.04)		(0.04)
Fixed effect	Yes	Yes	Yes	Yes
Constant	0.10	0.30	0.16	0.37
	(0.13)	(0.35)	(0.13)	(0.35)
Observations	3100	3100	3100	3100
R-squared	0.134	0.166	0.121	0.158

Note: Standard errors clutered at the class level are in parentheses. *** *p* < 0.001, ** *p* < 0.01, * *p* < 0.05.

**Table 5 ijerph-19-13317-t005:** PSM results analyzing the effect of self-esteem status on the academic achievement of students.

	Low Self-Esteem, 1 = Yes
1. Kernel matching	−0.130 ***
	(0.009)
2. Nearest neighbor matching	−0.154 *
	(0.021)
3. Radius matching	−0.169 ***
	(0.000)

Note: Standard errors clutered at the class level are in parentheses. *** *p* < 0.001, * *p* < 0.05.

## Data Availability

The data presented in this study are available on request from the corresponding author. The data are not publicly available due to it contains sensitive information on the mental health of children and adolescents in rural China.

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
