# Peer review of "The Role of Self-Esteem in the Academic Performance of Rural Students in China"

_ijerph, 2022, doi:10.3390/ijerph192013317_

Round 1
Reviewer 1 Report
The research presented is in line with the standards pursued in educational science. It yields interesting results on the relationship between self-esteem, contextual variables and academic results. It also provides the value of being conducted in a specific context, but investigated on the basis of reliable instruments and well-founded decisions. However, minor changes are needed for publication.
Introduction
Lines 86-113 contain aspects traditionally included in the method (instrument, analysis, etc.). It is suggested to include the justification of the study in the introduction, to include or remove the explanation in the method, and to end with the objectives of the analysis. A possible proposal to consider would be to include a first paragraph contextualising and justifying the study in the introduction that presents the research questions, separate the theoretical framework in another heading and conclude with the research objectives, which would serve as a link between the introduction and the method.
Method
Under method it is suggested to include a paragraph presenting the type of study conducted (paradigm, type of research, etc.).
Results
Table 1 presents the means accompanied by their standard deviations, but also measured in percentages. The reading of the table is therefore confusing, as we have tried to integrate the frequencies of the nominal variables. It is suggested that the table be reworked, as it is difficult to understand.
Table 2 needs a more complete name, what statistics are presented in the table?
Discussion and conclusions
In line 367 it is expressed that the results do not find causal relationships, wouldn't it be correlational? Correlation is not the same as causation. Line 408 acknowledges that this is not an experimental study.
The discussion would be enriched by a reflection on the practical implication of the results obtained.
In line 418 the reference is separated and should appear together [95,96].
Reviewer 2 Report
A relevant topic for educational psychology is analyzed, providing a novel orientation toward the rural population. According to the structure of the manuscript: (1) a clear description of the research problem is observed; (2) the empirical antecedents consider relevant studies on the subject; (3) a methodological strategy with sufficient empirical support is proposed to achieve the aim of the study; (4) the study findings are adequately discussed. However, there are important issues that should be addressed prior to publication.
(1) Although the findings show the importance of self-esteem for academic performance, it is suggested to deepen the discussion on the factors that facilitate the development of self-esteem in students, since it will support the development of arguments to propose future action.
(2) It is suggested to discuss the contributions of the findings to the teaching-learning process. In other words, discuss the practical implications of the findings in terms of the educational process.
(3) About 47% of the bibliographical references are from the last 10 years, so it is suggested to increase this percentage to have more current antecedents.
Round 2
Reviewer 2 Report
After reading the latest version of the manuscript, I was able to verify that the authors corrected the comments suggested above. Therefore, the manuscript is accepted for publication.